# A Structural Analysis of Economic Processes by the Use of "Energy Content Value"

**Hidekazu Aoki** [1],* **and Nobuo Kawamiya** [2]

1   Independent Researcher (Member of International Society of Ecological Ecomomics),
    601 Diapalace-Motomiyacho, Motomiyacho 4-83-2, Showa-ku, Nagoya 4660851, Japan
2   Independent Researcher, Fujimori 2-188, Meito-ku, Nagoya 4650026, Japan; eeforum2000@gmail.com
*   Correspondence: aoki.hidekazu@nifty.ne.jp; Tel.: +81-90-5871-4596

**Abstract:** Since currency is a means of exchange, it forms a standard of value. Modern economics uses the function of currency as a "general standard of value" and describes the economic processes in the terms of value by the quantity of money. The most notable indicator to symbolize this situation is "Gross Domestic Product" (GDP). However, GDP, in general, does not indicate the economic process as a whole. It is necessary to add intermediate inputs to the GDP in order to calculate the total amount of industrial output of a country, and this total industrial output can never realize without energy supply. In other words, the total industrial output of a country and the total amount of energy supplied (= consumed) constitute an inseparable relationship like the front and back of the coin. This research tries to propose an alternative evaluation indicator based on "energy", which is the most important economic substance and the indispensable requirement for all human activities. Regarding this issue, this study deals with the following problems. First, we refer to a dualistic composition which Soddy had adopted, from the beginning, to classify wealth, which is a product of economic processes, into two categories: one on a physical dimension (Wealth I · II) and the other on a psychological dimension (Virtual Wealth). Then, based on the dualistic framework proposed by Soddy, we present that we can evaluate the economic process, through a compound-eye view, on two dimensions: the physical dimension (energy flow) and the institutional dimension (money flow). Next, we focus on 'the economic process presented by monetary value' (A) and 'the energy quantity which effectually supports this process' (B). We have introduced how to project (B) onto the constituent elements of (A). Thus, we take a case study on Japan by use of dualistic aspects approach.

**Keywords:** wealth; virtual wealth; available energy (exergy); monetary evaluation; thermo-physical evaluation

## 1. Introduction

### 1.1. Background

Economic activities are inseparably correlated with energy conversion/consumption:

"The concept of the commonality of energy is critical to an understanding of how the economy really works. When we think of energy, we might well limit our thoughts to obvious forms of energy such as coal, oil, natural gas and electricity, but we need to understand that 'energy' is a much bigger concept than this. Human effort is energy, and that energy is in turn derived from the food that we eat, which itself is another form of energy. The nutritional content of food can be measured in calories (a unit of heat), and human labour can be quantified in watts, a unit more commonly used to measure electricity". [1] (pp. 11–12)

"Since currency is a means of exchange, it becomes a standard of value. Since it is a medium for exchange, currency comes to bear a general and exclusive function to serve as a standard of value to represent the relative values of commodities. Money can function as a value scale because it is a unit of calculation: this calculation unit can generally present the values of commodities". [2] (p. 314)

Modern economics uses the function of currency as a "general standard of value" and describes the economic processes in the term of value by the quantity of money.

The most symbolic indicator to express this situation is "Gross Domestic Product" (GDP).

However, "GDP influences our governance systems more than we realize. Politicians are rated on their GDP achievements, which can make or break elections. Countries that boast high rates of GDP growth get invited to join exclusive global clubs, from the G20 to the Organization for Economic Cooperation and Development (OECD), which includes the world's 'wealthiest' countries. Business leaders are expected to increase output because this is captured in GDP and counts towards national economic success. Investment choices are predicated on GDP's present and future projections, as the power of international credit ratings has demonstrated: countries, companies and other organizations can be irremediably broken by a negative rating motivated by low GDP growth" [3] (p. 15).

In other words, "GDP has become much more than a statistic: it has been the overarching parameter of success and a fundamental ordering principle at the global and national level, establishing the economic and political 'rules of the game'. This has been true for all varieties of socialism and capitalism in the twentieth century, as the consensus on GDP growth was eminently cross-ideological and, therefore, more pervasive than any other political ideology" [3] (p. 16).

GDP measures the value added to the products traded in the market in monetary quantity. For this reason, increasing market transactions are considered to be "growth".

However, GDPs, in general, do not indicate the economic process as a whole. It is necessary to add intermediate inputs to GDP in order to calculate the total amount of industrial output of a country, which will be discussed in detail below. This total industrial output can never realize without energy supply. In other words, the total industrial output of a country and the total amount of energy supplied (=consumed) constitute an inseparable relationship like the front and back of a coin.

*1.2. Issues to be Solved by This Research*

This research, in the above background, tries to propose an alternative evaluation indicator based on the "energy" which is the most important economic substance and an indispensable requirement for all human activities.

On the basis of this vision and motivation, this research tries to elucidate the following issues.

**A**    Economic processes expressed in monetary value is tightly correspondent with (B): the relevant energy supply effectually supporting (A); therefore, if we notice and collate (A)&(B) correspondence, our investigation on the development of a national economy will become more reliable and effective than a conventional one. Is it possible to verify the validity of this approach through real facts observed?

**B**    Considering the close relationship between (A) and (B), there is a possibility of a methodology that projects (A) components onto (B). Then, the research problem here is to apply this approach to a real economy and to verify its validity.

**C**    This approach means that economic activities be assessed by the relevant energy quantity in the dimension of (B), i.e., that a monetary evaluation can be replaced for an energy-quantitative evaluation. Application of this method (expression of economy in the dimension of energy) enables us to assess and analyze real economic activities in the dimension of Energy. When the results are considered valid and exact, this approach might form an alternative evaluation system in parallel with a currency-based one. The research problem here is how to practice this method to a real economy as explained in the next section.

**D**    By using the evaluation indices of (C), it becomes possible to compare economic performances direct over time and space without going through the time-point correction (deflator, discounted present value, etc.) and exchange rate adjustment in monetary value. The problem here is to verify the consideration that is applied to a real economic process.

## 2. Theoretical Foundation of This Research

### 2.1. Issues Raised by Nicholas Georgescu-Roegen

Nicholas Georgescu-Roegen wrote, in 1971, "*The Entropy Law and the Economic Process*" [2] and pointed out that the economic process follows "the entropy law, which says that the quality of matter/energy that gives rise to usefulness is used up and is not recyclable" [4] (p. 195). This indication demonstrated clearly that the "the circulating flow model" which constitutes the groundwork of the mainstream economics is a mere fiction, and therefore it cannot represent the true movement/transformation of goods in real-time and space. To put it simply, the existing economics considers the economic process to be a process of production/circulation of utility, whereas Georgescu-Roegen established an insight that it must be a process of entropy (ineffectiveness) increasing flow.

Thus, his entropy theoretic approach stands in the opposite of the value–utility approach in traditional economics. These two approaches can hardly be reconcilable, and the more he devotes himself to a thermodynamic approach, the lower the possibility that his approach can be understood from existing economics.

In other words, the more deeply Georgescu-Roegen's approach strengthens physical logic, the farther it will deviate from the traditional framework of economics as "production and circulation of utility". For this reason, his approach has become far isolated from the ordinary economic thought and is essentially unintelligible for the majority of economists.

However, the problem raised by Georgescu-Roegen had strong potential to bring about fundamental changes to economics. Accepting Georgescu-Roegen's approach, there appeared several economists who fully undertook his proposition that it is necessary to introduce a physical value scale in economy that is overwhelmingly monetary-based (cf. [4] pp. 192–193).

However, a theoretical perspective had already been given by Frederick Soddy (in 1926), enough to make a breakthrough in this problem.

Soddy, as one of the founders of nuclear chemistry, tried to apply the entropy theoretical analysis to the real national/world economy, and to contrast his result with the ordinary monetary approach in the traditional economics. Thus, Soddy put a scalpel of theoretical analysis on the discrepancy between the two approaches, the cause (origin) thereof, and the very extent of the discrepancy.

In this regard, Soddy's theory has revolutionary and precursory significance. On paying close attention to Soddy's unique point of view, we reconsider the theoretical framework originally conceived by Soddy in more detail in the following section.

Incidentally, Soddy gained an insight that existing economics are ignorant of the thermodynamic entity of the economic process, and therefore, which would inevitably cause a discrepancy (deviation) extant between the economic utility (value/price) and the physical usefulness. However, he jumped too far ahead: instead of *analyzing* the nature and structure of this discrepancy, he directly began to conceive an institutional mechanism to solve the divergent structure. Regrettably, this attempt of Soddy seems to have been too large a jump for most economists.

### 2.2. Theoretical Insight of Frederic Soddy

Preceding Georgescu-Roegen about a half century, Soddy adopted the entropy concept explicitly in the social context and struggled to apply this concept to the explanation of concrete economic processes.

First of all, Soddy defines the word "Available" as follows.

"The term available in this definition has the same meaning as in the second law of thermodynamics, which divides energy into two categories, useful, available or "free" energy, and useless, unavailable or "bound" energy, the latter also being designated entropy" [5] (pp. 108–109).

With such a premise, Soddy defined "The physical definition of wealth is a form or product of energy or work which enables or empowers life" [5] (p. 118).

They are described by the following mathematical expressions [5] (pp. 118–119).

$$Raw\ Materials + Available\ Energy = Wealth\ I. \tag{1}$$

$$Wealth\ I = Life\ energy + Waste\ Energy\ and\ Materials. \tag{2}$$

$$Raw\ Materials + Available\ Energy = Wealth\ II + Waste\ Energy. \tag{3}$$

These will be explained in detail by Soddy's words as follows.

(1) The production of Wealth as distinct from Debt obeys the physical laws of conservation and the exact reasoning of the physical sciences can be applied. Wealth cannot be produced without expenditure, and a continuous supply of wealth cannot be supplied as the result of any expenditure-once for all for it is a form of energy or the product of its expenditure under intelligent direction. Its production demands a continuous supply of fresh energy and continuous human diligence, nowadays, rather than physical labour. The scale on which it can be produced is practically limited only by the state of technical knowledge of the time. There is no longer any valid physical justification for the continuance of poverty. The phenomenon of unemployment and destitution at one and the same time to-day is solely due to ignorance of the nature of wealth and the principles of economics, and to the confusions between wealth and debt which have hitherto bemused that subject, even among those who have essayed its scientific investigation and elucidation.

(2) There are two distinct categories of wealth which owe their value to the opposite qualities of perishability and permanence. Both are alike in the manner of their production. But in the formation of the first category of perishable wealth the energy required is stored up for use later by life when the wealth is consumed. It includes food, fuel, explosives, fertilisers and an materials the usefulness of which depends upon the change they suffer in use. They can only be used once, and they usually function as the energisers and actual supporters of life.

In the second category of permanent wealth the energy required to produce it is not stored in the product—or, if it is, it acts detrimentally to durability in use—but has already gone to waste in the process. It enables and facilitates life, but does not empower it. It saves further expenditure of life-time to an indefinite extent but does not support life. The category includes all classes of permanent possessions, of all degrees of actual durability, but distinguished from the first category by the fact that their destruction is incidental to and not the reason for their usefulness, and is a dead loss. This category includes the whole of capital in the sense used in this book, namely, organs of production used in production" [5] (pp. 294–295).

To understand the essential concepts presented by Soddy, the following illustration will be of great use (Figure 1).

The "physical conservation law" mentioned by Soddy is, of course, the first law of thermodynamics: the law of energy conservation. Every kind of energy conversion follows the law of energy conservation. On occasion of using (converting) energy, "available energy" does not simply disappear but rather "converts itself" into "waste energy" such as waste heat, etc. However, ordinary people (non-physicists) would consider a "conversion" to be a real "extinction" because the "waste energy" is entirely inert, fully deprived of its usefulness (effectiveness) as energy: this type of energy is useless (unavailable) and meaningless for people, who would consequently regard such an energy as nonextant, or further, nonenergy; and as such, an "energy conversion" of energy as a simple "energy consumption".

Yet this view is false, according to Soddy, because the "Waste Energy" is a really existing entity (with its proper effects usually undesirable) inevitably produced by the energy conversion.

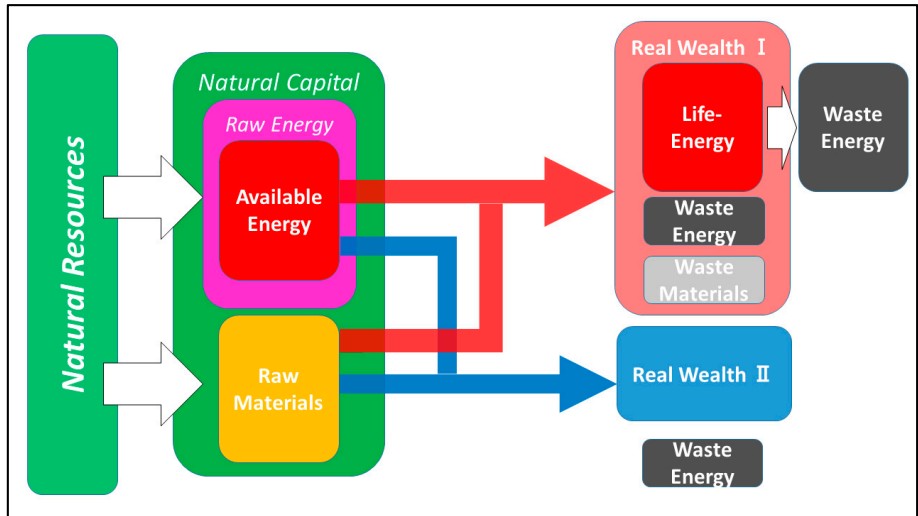

**Figure 1.** A detailed illustration of the process for the production of real wealth (originally presented by Soddy).

Then Soddy clarifies that there are two categories of wealth. (In the following, we will call these kinds of wealth "Real Wealth", in distinction of another important concept "Virtual Wealth" also presented by Soddy).

Real Wealth I consist of goods taken out of natural resources to cover the energy necessary for social life, such as food and fuel. This wealth is consumable and exerts its utility in the process of consumption (energy conversion). We acquire life energy from available energy and raw materials in natural capital and, at the same time, we emit waste energy and waste materials. The former and the latter are entirely inseparable, and the obtained life energy is also converted into waste energy.

In contrast to this substantial Wealth I, Real Wealth II is a medium (capital goods and durable consumer goods) useful directly and indirectly on using Real Wealth I: clothing, housing, machinery, etc. In the Real Wealth II, the available energy expended for processing raw materials is not retained in the result as an extant formation but is firmly "embodied" into the product itself.

Real Wealth I and Real Wealth II are both under the constraints of the conservation law.

Also, to obtain both, it is necessary to input Available Energy and Raw Materials. This is just what Soddy meant by saying that "wealth cannot be produced without expenditure."

However, once we acquire a certain real wealth, there will inevitably be left behind "waste energy and/or waste materials." The mathematical expression presented by Soddy meant that the 'production' and 'consumption' of real wealth strictly follow the second law of thermodynamics, i.e., the law of entropy.

### 2.3. Virtual Wealth and Debt

Another core of the Soddy theory lies in his essential insight that money is merely a "Virtual Wealth", which is essentially connected with Debts [5] (pp. 295–296).

Soddy presented money as a "dual concept" of 'Virtual Wealth and Debt'. This concept gets to the essence of credit creation and managed currency system that "creating money from debt" = debt money.

In addition, Soddy clearly pointed out that the virtual wealth of money has a faulty propensity to diverge disparately from the real wealth because the former, virtual wealth of money, does not directly link to the real wealth.

To prevent this fault, he advocated that (1) the government should be responsible for maintaining currency value and price stability, (2) the creation of bank credit should be prohibited, and (3) the currency should be put under the international and totally floating exchange rate system.

In the era of Soddy, the currency issuance by the monetary authorities was still linked to the real asset "Gold Reserve". For this reason, Soddy's claim was headed solely to the prohibition of excessive credit creation by commercial banks.

However, even though the monetary system was incomplete for Soddy, it was still working realistically and was generally accepted at that time.

However, Soddy did not carefully consider this circumstance but ardently insisted on his own sense of crisis: the divergence between Wealth and Virtual Wealth and its resultant danger and devoted himself to preach Monetary Reform enthusiastically. Thus, his stance here seems to have been his limitation. As a result, his thought was never understood in public, and he was handled as a mere eccentric.

*2.4. Contemporary Significance of Soddy's Theory Reconsidered*

Unlike energy, money never loses its value through one-time use: money holds its purchasing power on every time of use as far as its distribution is valid. Money makes it possible for people to repeat an economic act of purchasing "utility".

For example, the exchange equation $M \times V = P \times T$, which was formulated by Irving Fisher on the basis of monetary quantity theory, implies a premise that purchasing power of money can be reproduced any number of times. This formula shows that M (Money: money amount) multiplied by V (Velocity: circulation rate of money) matches, ex post facto T (Transaction: transaction amount of goods/service in one period) multiplied by P (Price: price) [6].

On the left side, M means amount of money and V means the turnover rate that shows how many times the money is used, so the units of this expression become a national currency such as US dollar, Euro, Japanese yen, etc. The money amount M can be regarded as the amount of currency called 'monetary base' or 'base money'. In short, it is the money which the central bank has created.

Thus, money has the essential "function" to reproduce this purchasing power perpetually. This seems to be the reason why the currency is generally regarded as universal wealth, and why the economics can describe the economic process as cyclical.

The right side requires a little caution. The expression $P \times T$ is a kind of collective concept, which means the sum total of the amount of sales of every goods (it is equal to the Transaction multiplied by its unit Price). Therefore, the unit of this expression is also of a national currency.

Thus, money has the essential "function" to reproduce this purchasing power perpetually.

In this respect, we think that Georgescu-Roegen's viewpoint of analysis did not cover such a function of money. This seems to be the main reason why Georgescu-Roegen failed to complete his theory to explain the economic process in a unified framework with the entropy concept.

In contrast to the entropy–monistic theoretical composition by Georgescu-Roegen, Soddy had adopted, from the beginning, a dualistic composition to classify wealth, which is a product of economic processes, into two categories: ones on a physical dimension (Wealth I · II) and others on a psychological dimension (Virtual Wealth). In addition, the physical definition of wealth was 'a form of energy or the product of its expenditure under intelligent direction'.

On the basis of the dualistic constitution first presented by Soddy, we can evaluate the economic process, through a compound-eye view on two dimensions: the institutional dimension (money flow) and the physical dimension (energy flow).

Here, we focus on 'the economic process presented by monetary value' (A) and 'the energy quantity which effactually supports this process' (B), and we have introduced a dual aspects approach for both (A) and (B).

## 3. Theory of Energy Content Value

### 3.1. Introduction of 'Energy Content Value' as a Value Concept

According to Soddy's physical definition of wealth as 'a form of energy or the product of its expenditure under intelligent direction' (i.e., the former is Wealth I and the latter is Wealth II).

This insight of Soddy seems to suggest that economic process can also be measured by the physical measure of energy content.

The prime subject of this research is to establish the methodology by which to realize Soddy's suggestion.

We would like to introduce a concept of 'Energy Content Value' for an analytical/comparative study of economic systems based on this concept.

But what does the Energy Content Value (ECV) mean?

Let us explain the concept of ECV by dealing with an instance in the electric power industry.

The distribution of electricity can be treated as a delivery process of power product (commodity) to consumers through facilities for generation, transformation, and power services; meanwhile, the process is, in its physical substance, to extract available energy in the form of electric power from primary energy like hydropower, coal, petroleum, natural gas, uranium, or other natural energy resources.

Thus, there can be two different and complementary viewpoints from which to deal with economic processes:

One is from the monetary value basis, which regards the economic process as a successive addition of values (utility). The other is from the real quantity basis of the relevant physical substances, which regards the said process as successive conversions of primary energy into available energy (*exergy*) and unavailable energy (*anergy*).

In the recent Japan, the power has a price of ¥21 per kWh on the average over home use and industrial demands: this price represents an economic value of the energy delivered to the consumers.

On the other hand, the energy amount represented by "kWh" (the trading unit of electricity) is only a part of the total primary energy input into this electricity. Of course, this total primary energy does not actually (as a physical quantity) exist in the product: for example, the former includes the relevant energy input in the past (before the current power service).

However, when we refer to the "value/price", it becomes possible to understand the complex situation more clearly. Effectually the consumers purchase, by paying ¥21 per kWh, "Value of Energy Content" corresponding to this overall primary energy.

Focusing on this kind of relation, we try to introduce an alternative value concept called "Energy Content Value". Here, we would like to define that it means "an evaluation of a certain good or economic activity in reference to the amount of all the primary energy inputs it requires." We redefine it exactly in Section 3.3.

### 3.2. Primary Energy Supply (PES) and Total Industrial Output (TIO)

Naturally, the 'price' is not attached to energy resources or to raw material resources at the initial stage: they acquire the prices just when they are converted into "Life energy" or "Real Wealth II" through the production process shown in Figure 1 and become 'tradable commodities'.

The price consists fundamentally of the costs required to cover the labor force and the capital equipment put into the production process. Entrepreneurs will add their profit to these costs and send thus price the commodity on the market. Of course, under the market economy, prices are not valid without acceptance by consumers. Therefore, commodity prices in general distribution in the market are considered to reflect the expenses (and profits) spent in the production process.

We already have annual macroscopic data called the Input = Output Table, which collects commodity prices in each industry.

In the data, the intermediate input (equal to intermediate consumption), employee income, operating surplus, capital depreciation, final consumption, capital formation, etc. are estimated in matrix form in vertical and horizontal directions.

In the input–output table, the count sequence in the vertical direction is called 'column'. Each column shows the breakdown of the payment to raw materials, fuel, labor force, etc. used in the production of goods and services of each department (cost structure), which is called "input" in the input–output table.

On the other hand, the count sequence in the horizontal direction is called 'row'. Each row shows the breakdown of the sales destinations of goods and services produced in each department (sales channel composition), which is called "output". [7]

'Total of Gross Value Added Sectors', which calculates the total of 'Employee income', 'operating surplus', 'capital depreciation', and 'indirect tax (less) current subsidies', shown in the 'column', is consistent with 'Gross National Income (=Distributed GDP)'.

'Total Final Demands' (which calculates the total of 'Consumption expenditure', 'gross domestic fixed capital formation', 'increase in stock' and 'export') minus 'Import' shown in the 'row' is consistent with 'Gross Domestic Expenditure (=expenditure GDP)'.

On the input–output table, the sum of the final demand and intermediate input is called "Demand total"; in this research however, it is called "Total Industrial Output" from the viewpoint of the production side.

On the other hand, the production process in Figure 1 overlaps with Primary Energy Supply (=consumption). PES is usually displayed in "Energy Balance Table" format. Here, we come to deal with this issue.

The input–output table is actively used for measurement of "economic ripple effect" and the like. In the field of environmental economics, it is applied to the estimation of energy consumption and carbon dioxide emissions by industry, etc. In Japan, the National Institute for Environmental Studies, Center for Global Environmental Research (CGER) has compiled and published 'Embodied Energy and Emission Intensity Data for Japan Using Input–output Tables (3 EID)' [8].

In this connection, the Institute of Energy Economics, Japan conducted 'Estimation of energy input table by industry sector to be available for the economic, industrial, energy, and environmental analyses.' The result of this estimation was obtained on the following method. (a) To use the input monetary based ratio on the SNA input–output table and (b) to combine this with the domestic demand total (=input from conversion section + final demand total) for each energy source obtained from the energy balance table [9].

These estimates are seeking for 'real quantities', such as the energy input or carbon dioxide emission. These estimates would be sufficient as far as we estimate the input effective energy or the physical quantity of substances to be finally diffused as entropy.

This treatment for the energy estimation, to be noted, includes the energy inputs to produce social capital, production equipment, and durable consumer goods that belong to the category of 'Soddy's Real Wealth II', without any distinction. However, it is more reasonable to assume that these energy inputs are '*embodied*' into the products (cf. Section 2.2. and Figure 1), although these inputs have completely been *dissipated* in the form of energy.

We can grasp energy inputs to specific products or business activities by Inventory Analysis used in Life Cycle Assessment. Nonetheless, this approach is hardly applicable to extremely large-scale systems like a series of social capitals. In fact, there seems to have been no such attempt.

Therefore, we focus on the relationship between Total Industrial Output (TIO) obtained from the input–output table and Primary Energy Supply (PES) and try to organize TIO/PES correlation theoretically as follows.

We have accurate annual data on primary energy supply (freely available) assembled for a long term. Furthermore, from the flow chart on annual primary energy supply, we have been able to discern the energy used as available energy from the one disposed as waste energy.

The PES expresses the total energy used for all the economic production per year. On the other hand, the TIO expresses the total monetary value of economic production per year.

From the above-mentioned metrics, the two macroscopic statistics <PES and TIO> are considered to be the two aspects of the same economic entity. In other words, Primary Energy Supply (PES) depicts the total economic activity from the perspective of energy. On the other hand, Total Industrial Output (TIO) expresses the same process by currency valuation.

### 3.3. Exactly Definition of "Energy Content Value"

There is a close correspondence between "Total Industrial Output (TIO)" = "the economic activities represented in the monetary value", (A), and "Primary Energy Supply (PES)" = "the energy amount that effectually supports (A)", (B).

Thus, we try to propose an evaluation/analysis framework that evaluates the overall economic activity with "Energy Content Value" by projecting constituent elements of TIO (A) onto the PES (B).

This evaluation/analysis framework consists of principle and assumption as follows.

I　　Principle: Correspondence between TIO and PES as a whole: Any Industrial Output (IO) requires some relevant Energy Supply (ES) and, vice versa, any ES produces some relevant IO. Therefore, TIO and PES necessarily correspond as a whole, even if correspondence between the parts of TIO and the parts of PES is not uniform (incomplete).

II　　Assumption: Correspondence between the TIO components and the relevant PES components Unlike the abovesaid case, an automatic correspondence is not necessarily secured in this case; however, we can make the two sides correspond mutually by introducing 'the principle of correspondence in detail', in which each TIO component should have an 'Energy Content' = PES × TIO component (=each IO value)/TIO. The individual ES term thus obtained is a physical quantity having the energy dimension (TOE or kWh) and, at the same time, a value quantity expressing the relevant IO exactly.

III　　Definition: In fact, the Energy Content Value (ECV) means "the value amount of an economic activity in the energy dimension derived from the principle ① and the assumption ②".

The above is shown in Figure 2.

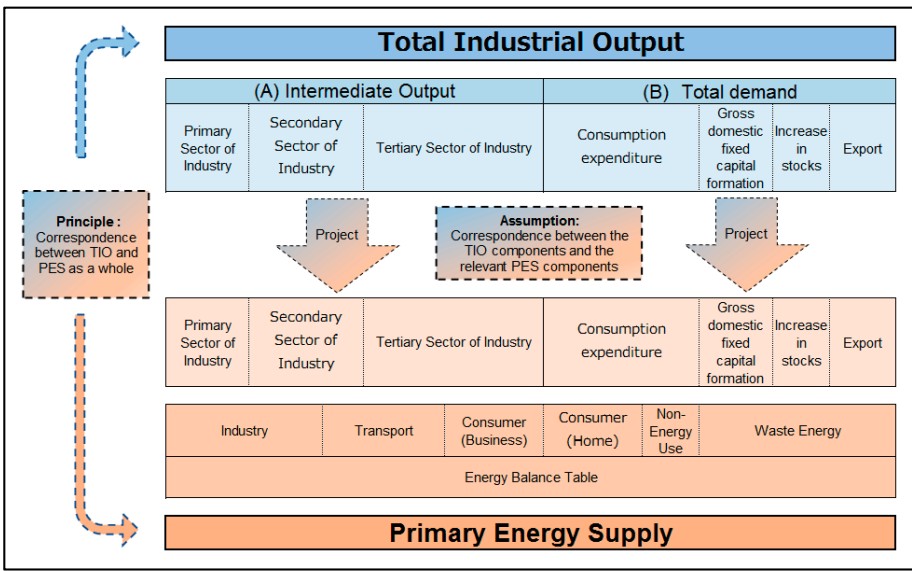

**Figure 2.** Exact definition of "Energy Content Value" in this study.

The quantity prescribed here was a 'projection of a TIO expressed in the monetary value basis' onto the PES coordinate.

In this projection, the prototype is in the monetary value dimension, and its image is in the energy dimension.

For example, Japan's TIO was 1101.1 Trillion Yen in 2015 when calculated by the method already described in Figure 3 [10]. On the other hand, the PES of Japan was 502.3 Million tons in the same year. [11] Therefore, the conversion coefficient from the TIO to the PES was 502.3 ÷ 1101.1 = 0.456 [TOE/Million Yen] in 2015.

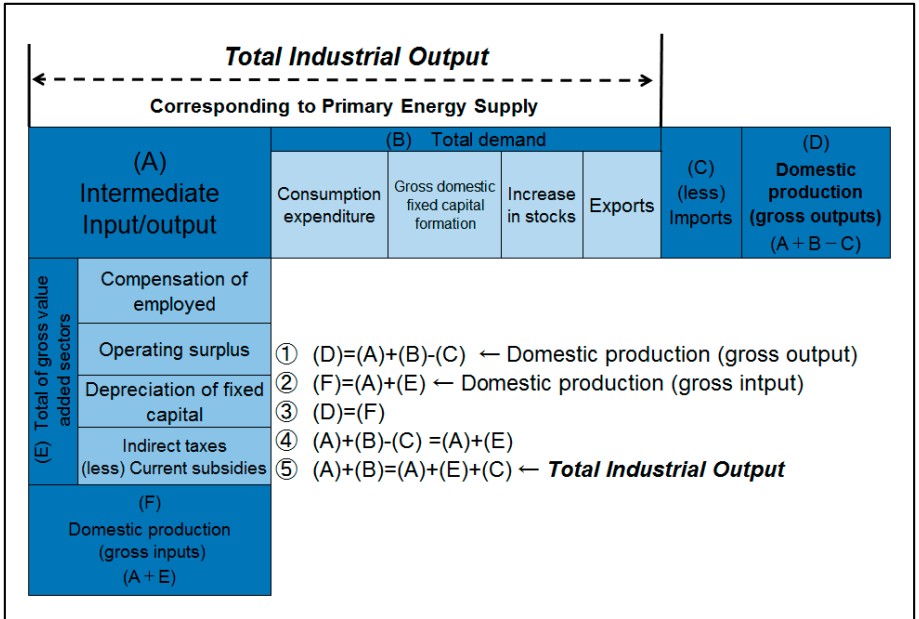

**Figure 3.** The conceptual frame of the input–output table. Source: The authors' interpretation on the general inter-industrial input/output table.

If we apply this proportionality factor to the output of each industrial sector, then we obtain an image of the output in the dimension of energy.

In proportional expression:

$$1101.1[\text{Trillion Yen}]:502.3[\text{Million TOE}] = 1[\text{Million Yen}]:0.456[\text{TOE}] \tag{4}$$

=an industrial output based on the monetary value [in Million Yen] vs. the relevant output based on the ECV [in TOE].

The last term in this proportional equation signifies the relevant industrial output in the dimension of energy.

To use this proportionality enables us to convert the industrial outputs expressed on the monetary basis to those expressed on the energy basis (dimension).

As a result, it has become possible to directly collate the industrial outputs as projected onto the energy coordinate and the real amounts of energy consumed by the relevant industries because the former and the latter are on the same coordinate (i.e., the dimension of energy).

Such a comparison might have been impossible as far as the industrial outputs are measured solely on the monetary basis.

What kind of effects does this collation have? Let us make a case study dealing with the Japanese power industry.

As for Japan's power industry, its PES in FY 2015 was 502.3 MTOE, of which, the electric power industry received 170.1 MTOE (33.8%) to provide the electricity of 71.5 MTOE (14.2%). [11].

On the other hand, Japan's Total Industrial Output of the same year was 1101.1 trillion yen of which the power industry received 19.8 trillion yen (1.8% of TIO). [10].

Such a simple juxtaposition of energy-based trades and monetary-based trades cannot elucidate the structural relationship of the two transactions.

However, if we apply the ECV concept to the deal of the electric industry, then we can approach the problem of structural analysis.

In short, Japan's electricity industry provides the economy with 71.5 MTOE of a real (physical) energy, while receiving only 9.0 MTOE of the ECV.

What does this situation mean? The power industry is considered to have donated the difference 62.5 (=71.5–9.0) MTOE to the social economy.

If the power industry confiscates all the ECV equal to the real quantity, there may be no ECV left available for other industries and households.

Tim Morgan says, "Ultimately, the economy is, and always has been, a surplus energy equation" [1] (p. 11).

In this connection, we have introduced an alternative value-scale ECV which enables us to quantitatively analyze the structural correlation between energy flow and economic activities.

### 3.4. Importance of Estimating the Available Energy Embodied into Real Wealth II

The essence of the Soddy theory is to describe and analyze, explicitly the fact that "the economic processes of mankind are all energy conversion processes".

In a sense, Soddy's theory has been realized (to a significant extent) by the statistical data on the Primary Energy Supply (PES). However, there is a serious limit to this kind of data: the data shows only the paths of energy flow but does not present how much available energy is embodied (materialized) in each item of Real Wealth II. It is necessary to calculate an amount of available energy realized in each part of the Real Wealth II. When this be done, it becomes possible to quantitatively grasp the total aspect of available energy flow in the economy.

On the other hand, "the total fixed capital formation", which is the Real Wealth II produced annually, can be grasped by the statistical representation of the Total Industrial Output (TIO) in the monetary (quantitative) expression. In other words, if we adopt a methodology to "match (collate) the two statistics of PES and TIO", we can attain a possibility to make a reasonable estimation of the effective amount of energy that was embodied in Real Wealth II.

However, what we estimate here is 'Content Value' when converted to primary energy amount. We would like to emphasize once again that it is not the 'real quantity' of actually input energy.

Energy resources that make up the mainstay of Real Wealth I have "flow-type resources" such as agriculture, forestry, fishery products, hydropower, wind power, geothermal, and solar heat, and "stock-type resources" of fossil fuels. The most fundamental significance of the industrial revolution was that we switched most of the energy resources from flow-type resources to much more abundant stock-type resources. Nonetheless, stock-type resources can never be reused once they are used. Therefore, since human beings decided to rely on this to live a social life, we have been destined to endure a fate of "depletion of resources".

Soddy was fully aware of this situation. In fact, he wrote: "The flamboyant era (from the end of the 19th century to the beginning of the 20th century) through which we have been passing is due not to our own merits, but to our having inherited accumulations of solar energy from the carboniferous era, so that life for once has been able to live beyond its income" [5] (p. 30).

In 1972, just one year after Georgescu-Roegen published his great work, the report of the Roma club entitled "Limits to Growth" was published. Since then, a strong sense of crisis was formed about the sustainability of Real Wealth I. In contrast, the problem of the sustainability of Real Wealth II has long been neglected entirely. The lifetimes of now extant Real Wealth II are considered bound to run out far before the point when Real Wealth I will diminish its sustainability.

Unless the company's production facilities are renewed, production capacity will be directly reduced. If this happens across all enterprises, employments and incomes will be lost at once and the national economy will soon be atrophied. If the nationwide social capitals, such as the power

network, the railway network, the road network, etc., be not updated properly, social life will fall into great disorder.

Daily pointed out that "the curious asymmetry in present national accounts whereby we write off the value of man-made assets against current production as they depreciate but make no such deduction for depreciation of natural assets" [3] (pp. 196–197). Although this indication is certainly important, in the first place, 'depreciation' of fixed assets is only an accounting technique of to divest upfront investment as part of expenses to collect from subsequent income. Actually, depreciation by a fixed amount or fixed rate is merely an accounting method that is prescribed without considering physical deterioration of Real Wealth II.

More specifically, even if the depreciation of a certain commodity was properly implemented, yet the reserve might not always be possible to update the commodity at the end of its life. Although it may be certain that the investment of the past be collected safely, there is no guarantee that the reinvestment be applied, without failure, to the renewal, and there is also a high risk that the necessary renewal costs rise steeply in price at the time of renewal.

But, to replace the monetary value by the Energy Content Value will make it possible (A) to compare directly the economies in different time points and (B) to do that without use of the discounted present value or of the deflator for "point fix" and, in addition, (C) to estimate an amount of fixed capitals to be required for the updating and/or maintenance in the future on the basis of Energy Content Value. Also, (D) an international economic comparison will become possible direct without interference of the foreign exchange rate.

It seems that this methodological approach has a wide promising applicability.

In the next section, let us take a case study on Japan based on this approach.

## 4. Trends of PES, TIO, and GDP and Evaluation of Railway Construction Project by ECV: A Case Study in Japan

### 4.1. Trend of PES and TIO, GDP in Japan

Let us survey the historical trends of Total Primary Energy Supply, Total Primary Energy Supply, and Gross Domestic Product in Japan as represented in Figure 4.

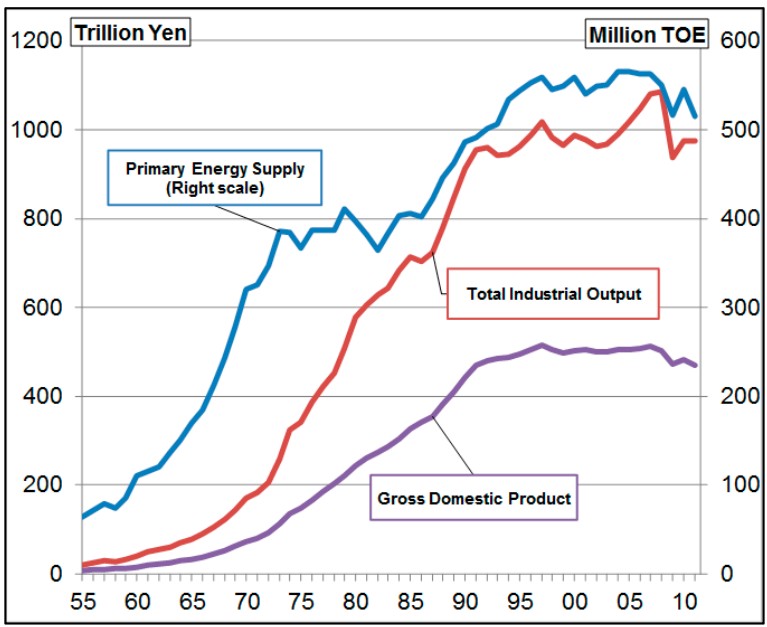

**Figure 4.** Total Primary Energy Supply, Total Industrial Output, and Gross Domestic Product of Japan, 1955–2011.

First, let us see the trends along the horizontal time axis.

The high economic growth in Japan has simply been regarded as a rapid increase in the GDP. It should be understood, however, in its historical perspective.

On 15 August 1945, the Asian-Pacific War reached the end by Japan's unconditional surrender. The industrial section of Japan had been practically devastated due to the heavy air strikes by the US Air Force.

For the postwar reconstruction, the Japanese government first conceived the 'Priority Production System': "The idea is to raise coal and steel production in a cross-cyclical manner through imported heavy oil—increasing steel production—priority distribution of steel materials to coal mines—increase coal production—priority distribution of coal to steel, and to try to stop reducing re-production" [12] (p. 4).

The Government of Japan set a goal to coal production of 30 million tons in 1947. To realize this output, the Government expected, as important leverage, emergency import of 80,000 kl of heavy oil through the Occupation Army and, in fact, "The timing when production began to show a steady increase was after the emergency import of heavy fuel oil was started by US aid" [13].

As for the outcome of the Priority Production System, it should be recognized as fact that the expansion of energy supply was a prerequisite for Japan's postwar reconstruction although there are various negative views on the PPS policy.

Since then, the Japanese Government established the energy supply system by "Power Development Policy" with the mainstay of large-scale hydroelectric power generation. There was such a preliminary stage during the rapid economic growth of Japan.

Then, Japan's Primary Energy Supply grew by an extraordinary ca 6.5 times (equal to 9 percent per year) or from ca. 60 million TOES (Tons of Oil Equivalent) to ca. 400 million TOE in the two decades from the early 1950s to 1973—the year of the First Oil Crisis. Primary Energy Supply then (in 1973) suddenly halted at ~400 million TOES and went into a temporary plateau. From 1983 it resumed a slow expansion, reaching a second plateau of ~500 million TOES from mid of 1990s.

Meanwhile, the TIO of Japan (in terms of nominal yen) began to make a very rapid rise with a time lag of ca. ten years to the PES rising. We think that this delay is an indication that the rise of capital investment preceded that of TIO/GDP. We analyzed that energy demand expanded first, in need for capital investment, then production facilities expanded, leading to a production increase, which finally enlarged GDP/TIO.

Since the first oil shock in 1973, the PES did not regain its former rising momentum. On the contrary, TIO had survived this oil shock and successfully continued to rise for ca. twenty years thereafter. On the background, we think that there was an expansion of productivity by former capital investment, and fixed capital investment in the bubble period. However, the expansion of TIO had completely ceased in the early 1990s and, thus, economic growth brought by the expansion of PES and TIO became impossible.

In addition, Japan "is entering an uncharted societal stage of perpetual population declining, after achieving the world's longest life expectancy" [14].

According to the latest estimate by the National Institute of Population and Social Security Research (estimated in 2017), the total population of Japan will decrease to 119.125 million in 2030. It will be a decrease of 8.959 million from the peak of 128.084 million in 2008. This reduction is comparable to the population of Austria in 2017: 8.773 million people. Further, Japan's population is estimated to decline to 101.923 million in 2050 [15].

In other words, from the peak time to 2050, a population corresponding to those of Australia (24.7 million people) and New Zealand (4.794 million people) in concert in 2017 will disappear from Japan.

Population declining in Japan is due to a progress of decreasing birthrate. The birthrate in Japan is greatly below the population replacement level. The population composition is extremely weighted

to the senior side, and the population of labor age is decreasing steadily. In this situation, it seems inevitable to think that economic growth potential be affected by population decrease.

Next, let us trace the trends vertically to the time axis. This will contrast PES and TIO along the time span.

We can get the ratio of TIO/PES or PES/TIO, which gives, respectively, a unit cost or a unit yield of Energy Content Value for the national economy. This is shown in the following Table 1.

**Table 1.** Historical Shifts in Japan's Primary Energy Supply, Total Industrial Output, Gross Domestic Product (Numerical Resume), and Conversion Coefficient.

| Item | | Unit | 1955 | 1965 | 1973 | 1985 | 1990 | 2000 | 2011 |
|---|---|---|---|---|---|---|---|---|---|
| Primary Energy Supply | | Million TOE | 64 | 169 | 385 | 405 | 486 | 559 | 516 |
| Total Industrial Output | | Trillion Yen | 21 | 77 | 258 | 713 | 912 | 987 | 976 |
| Gross Domestic Product | | Trillion Yen | 8 | 33 | 112 | 325 | 443 | 503 | 471 |
| Conversion Coefficient | PES to TIO | Million Yen/TOE | 0.32 | 0.45 | 0.67 | 1.76 | 1.87 | 1.77 | 1.89 |
| | TIO to PES | TOE/Million Yen | 3.10 | 2.20 | 1.49 | 0.57 | 0.53 | 0.57 | 0.53 |

*4.2. Evaluation of Railway Construction Project by ECV*

We are now trying to evaluate the projects of Linear Shinkansen, Tokaido Shinkansen, Linimo (the only linear railway in Japan) by ECV, [16] which will elucidate the details; here, however, we will show the brief results.

First, we outline the three projects.

The Linear Shinkansen plan is a project to connect Tokyo-Osaka (438 kms) in 67 min by magnetically levitated (superconducting maglev.) bullet trains, with a maximum speed of 505 km/h at a total cost estimated at 9.03 trillion yen. The first stage object is to start service on the Tokyo-Nagoya 286 kms sector by 2027, for an investment of 5.43 trillion yen. Overall completion of the line through to Osaka, under the revised Abe government plan, is to occur in 2037.

The Tokaido Shinkansen opened on 1 October 1964, just 10 days before the opening ceremony of the Tokyo Olympics. This line, well-known as a global trailblazer of the high-performance intercity express, has a span of 515 kms. The total construction cost was 380 billion yen, of which, 28.8 billion yen (80 million dollars) was supplied by the World Bank (at the then exchange rate of: 1 USD = 360 Yen).

Japan has now one functioning linear railway, called LINIMO, i.e., the Aichi High-Speed Transit Tobu Kyuryo Line built to service the 2005 Nagoya Expo. Its trains are magnetically levitated and driven by normal-conductive (not superconductive) coils. This tiny (9 km) line cost 100.8 billion yen, nearly equivalent to 900 million dollars (at the 2005 exchange rate of 110/USD). Astonishingly, its cost was roughly equal to the total cost of Tokaido Shinkansen (approximately 1.06 billion dollars when calculated at the 1960s rate of 360yen/USD).

The cost comparison of the monetary value display of these three projects is shown in the table as follows (cf. Table 2).

**Table 2.** Comparison of the Linear Shinkansen and the two previous railway projects.

| | Item | Unit | LMS (Plan) | Tokaido Shinkansen | Linimo |
|---|---|---|---|---|---|
| I | Starting Service | Year | 2045 | 1964 | 2005 |
| II | Route Span | km | 438 | 515 | 9 |
| III | Maximum Speed | Km/h | 505 | 285 | 100 |
| IV | Total Cost | Billion Yen | 9030 | 380 | 100 |
| V | CPI based on the averege price for 1934-1936 | | 1786.1 | 413.3 | 1765.8 |
| VI | Relative CPI (Consumer Price Index) | | 1.00 | 4.32 | 1.01 |
| VII | Total Cost in Current Value (IV × VI) | Billion Yen | 9030 | 1642 | 101 |
| VIII | Cost per 1km Railway (VII ÷ II) | Billion Yen/km | 21 | 3 | 11 |
| IX | Yen-Dollar Exchange Rate | Yen/$ | 105 | 360 | 110 |
| X | Total Cost on USD Basis (IV ÷ IX) | Billion $ | 86.1 | 1.1 | 0.9 |
| XI | USD Based Total Cost (Current Value) (VII ÷ 104.85 Yen/$) | Billion $ | 86.1 | 15.7 | 1.0 |

As already shown in Table 1, the total cost of the Tokaido Shinkansen would be 1.6422 trillion yen if rendered into the current price on the basis of the CPI (consumer price index). The Linear Shinkansen is estimated to cost at least 9.03 trillion yen, ~ 5.5 times more than the Tokaido Shinkansen. If we use the CGPI (corporate goods price index) with 2010 as 100, the Tokaido Shinkansen's original (1964) cost of 380 billion yen will become 803.4 billion yen, which is yet substantially lower than the CPI-based conversion, and the Linear Shinkansen's cost is 11.3 times greater.

The fact is that the Consumer Price Index (CPI) is about twice as large as the Corporate Goods Price Index (CGPI): this means that there can occur a serious inconsistency in estimates of the current cost/value. Yet we cannot find any "a priori correct" criterion on which to choose.

Finally, the cost comparison by ECV expression is shown. (cf. Table 3)

A former technical analysis has already shown that the Linear Shinkansen constitutes an abnormally energy wasting (low efficiency) project, consuming in operation between four and five times as much power as the Tokaido Shinkansen [17]. It is also the case that at the construction phase prior to operation, the Linear Shinkansen will consume seven times as much Energy Content Value (=10,893 TOE ÷ 1562 TOE) as the Tokaido Shinkansen even at the stage of construction; the provisional cost estimate for the Linear Shinkansen might have been set as low as possible, since the costs of such project budgets are conventionally to be underestimated.

Thus, the Linear Shinkansen turns out to be a project of unprecedented absurdity, massively wasting energy even at the construction stage alone.

**Table 3.** Energy Content Value comparison between the three railway projects of Linear Shinkansen, Tokaido Shinkansen, and LINIMO.

|   | Item | Unit | LMS (Plan) | Tokaido Shinkansen | Linimo |
|---|------|------|-----------|--------------------|--------|
| I | Starting Service | Year | 2045 | 1964 | 2005 |
| II | Route Span | km | 438 | 515 | 9 |
| III | Maximum Speed | Km/h | 505 | 285 | 100 |
| IV | Total Cost | Billion Yen | 9030 | 380 | 100 |
| V | Total Industrial Output | Trillion Yen | 975.7672 | 71.1374 | 1017.0229 |
| VI | Primary Rnergy Supply | Million TOE | 515.541 | 150.608 | 565.087 |
| VII | Conversion Coefficient (V ÷ VI) | Million Yen/TOE | 1.89 | 0.47 | 1.80 |
| VIII | Energy Content Value (IV ÷ VII) | Million TOE | 4.77 | 0.80 | 0.06 |
| IX | ECV Cost per 1km (VIII ÷ II) | Thousand TOE/km | 10.893 | 1.562 | 6.224 |

## 5. In Conclusions

In this research, we proposed an evaluation/analysis framework that evaluates the overall economic activity with "Energy Content Value" by matching the price of goods with the amount of Primary Energy Supply to the economy.

Specifically, we have correlated the Primary Energy Supply (PES) to Total Industrial Output (TIO) described on the input–output table.

The relationship between the two is as follows.

The Overall Balance: the TIO and PES are mutually correspondent in total in the respective economy unit. This correspondence is complete in principle: the PES fully flows into the TIO and no part of TIO is achievable without proper use of energy.

The Detailed Balances: the respective correspondence between a part of TIO and a part of PES is not self-evident in contrast to the above said balance in total but is theory-dependent. This research has proposed, as discussed in detail in Section 3.3, the most appropriate method possible (now available) for the correspondence (attribution) between the parts of TIO and PES.

In this research, we have proposed (1) to establish the 'concept' of 'Energy Content Value (ECV)', (2) to match (collate) the two statistics of 'primary energy supply (PES)' and 'Total Industrial output (TIO)', (3) to assign the parts of PES to those of TIO, and (4) to introduce a method to analyze economic processes from energy content aspect.

Here, it can be confirmed that GDP can hardly represent all the domestic economic processes. If we want to catch the whole economic process, we have to add intermediate inputs to GDP.

Moreover, by introducing this alternative approach, it becomes possible to evaluate the economic processes from the dual aspects of monetary value and energy content value.

As a practice of viewing from both sides, we presented a case study that compared the historical trends of Japanese PES and TIO on the same stage in Section 4.1. In this connection, general energy statistics shows energy demands and their GDP elasticity. However, do not overlook that it is not GDP but TIO that corresponds directly to PES.

Furthermore, this approach will enable us to grasp the energy that has entered the production of fixed capitals or durable consumer goods and to assort that kind of energy as "embodied" into durable goods. Such an assortment has been unavailable in the traditional GDP approach, which treats energy as a uniform flow.

In addition, as indicated in Section 3.3, replacing the economic value standard from the monetary value to the energy content value allowed us to practice the following analyses.

(A)  To make a direct comparison between some economies in different time points.
(B)  To conduct (A) without use of the *discounted present value* or of the *deflator* for "point fix".
(C)  To estimate an amount of fixed capitals to be required for updating and/or maintenance in the future in terms of Energy Content Value.
(D)  To make an international economic comparison direct without interference of the foreign exchange rate.

Section 4.2 shows the specific example of the ECV-based comparison between three railway projects in Japan. As a result, this comparison is highly concrete; more so than that based on monetary value evaluation.

By using of the concept of ECV, the current money cost required for the construction of the Tokaido Shinkansen can be obtained as follows.

$$\text{ECV cost (TOE/km)} \times \text{Total Extension (km)} \times \text{ECV unit price as of 2011 (Million Yen/TOE)} = \text{Current Value of Construction Cost} \tag{5}$$

$$1562 \text{ (TOE/km)} \times 515 \text{ (km)} \times 1.89 \text{ (Million Yen/TOE)} = 1520.4 \text{ (Billion Yen)} \tag{6}$$

The comparison based on ECV here was that of domestic railway business beyond the time point. Further application is promising; for example, if you investigate the ECV unit price at a certain point in a foreign country, such as China or India, and grasp the total project cost and the total extension of railway investment at the proper time point, this will enable a direct comparison of the cost per km on the ECV dimension.

There has long been criticism to ecological economics that it gives only "qualitative (tentative) but not quantitative" arguments. This research will open a way to quantification.

As confirmed before, the mission of the energy industry or its raison d'etre is to provide society with a thermodynamic surplus.

On the other hand, nuclear power plants can provide energy surplus only for the period of power generation but, after their decommission, they will need an almost infinite energy forever in order to isolate nuclear wastes generated during the years of operation.

Such a burden may well exceed the surplus energy offered by nuclear generation; besides, it is the current generation that enjoys the energy surplus while the future generations are forced to devote their energy to contain nuclear materials.

In fact, the nuclear generation cannot stand without intergenerational exploitation.

Current generations having received nuclear power, in any country, are ethically required to accumulate "energy preparation" available for future generations.

This research will open up a clue to clarify the structure and the need for this preparation.

In this research, we presented a method to estimate the energy quantitative value of the output level by projecting the input–output table data onto PES.

The evaluation method proposed in this research is to reassess the production and the added values by each industrial sector using "Energy Content Values" based on the share of the monetary value criteria (occupancy weight) in the national economy.

In ordinary economics understanding, energy productivity means the amount of sales or profit against input energy. If the ratio is large, energy productivity is considered high.

This contrast structure is inherited also in this research. For this reason, if the sales amount is large as compared with the energy consumed, it might also give a result as if the input/output efficiency in the energy level be high.

This situation may seem to restrict the principle of this research.

However, the energy recognized as input or consumption in the traditional economics is completely limited to the "flow-level energy" (indicated in the energy balance table). On the other hand, the approach of this research can deal with quantitative evaluation of the energy "embodied" to production facilities, houses, and durable consumer goods (i.e., Wealth II of Soddy).

The total value of energy actually injected or consumed must be the energy quantitative value required for the operation of the asset (A) plus the energy quantitative value built into these fixed assets (B). It is {(A) + (B)} that should be dealt with for the truly effective energy analysis.

The approach of this research to estimate {(A) + (B)} for each power generation sector; is also attempted in another study.

This type of analysis became possible because the physical output of the electric industry is unambiguously clearly energy as shown in the case study of Section 3.3. Likewise, this type of analysis is applicable to city gas or similar energy supply industries. On the other hand, however, it is extremely difficult to apply this analysis for the general industries other than energy related industries.

Yet, energy-concerned efficiency of economic systems will gain ever increasing importance because resource constraints and environmental restrictions will become far severer in the future: the analytic approach of this research will make a basic and essential contribution to treat these problems without fail, although it will require further development of research.

Furthermore, if it becomes possible to assign an appropriate degradation process to Wealth II evaluated by Energy Content Value, it will enable us to estimate the physically useful life of immobile assets. Thus, the replacement costs of urban infrastructures can be estimated almost exactly because (I) input Energy Content Values of fixed assets are stored unchanged and (II) the current monetary values (at their replacement times) can be obtained as the products of (I) times (III): the conversion rates from ECV to monetary values. (II = I $\times$ III).

Such estimation will have a very significant meaning in nations like Japan that have invested enormous social capital investment in the past.

One of our next tasks will be to establish a methodology to carry out such estimations concretely.

Although the above-mentioned example confirms the efficacy of this method, the most essential advantage of this study lies in the establishment of the very concept of "Energy Content Value".

From now on, Japan is concerned about its declining population and the resultant economic contraction, which might be supplemented with growth of per capita GDP by introducing robot technology, artificial intelligence, automatic driving of cars, more advanced IoT, etc.

However, the economy would not be sustainable if the very energy consumption expands as a result of too heavy technical equipment.

Evaluation of economic activity from the energy content aspect will become even more necessary.

This research has advocated an introduction of an alternative value concept, the "Energy Content Value", which could work as an effective evaluation criterion for a present and future economy and might make a forerunning work in this new perspective.

In search of a more reliable value scale, we introduced an alternative value concept called "Energy Content Value", and we explored ways for reasonably dealing with the real (not virtual) economic system. Of course, the approach presented in this research should further be verified empirically in accordance with real economic processes.

**Author Contributions:** Conceptualization, H.A.; Data curation, H.A.; Formal analysis, H.A.; Funding acquisition, H.A.; Investigation, H.A.; Methodology, H.A.; Project administration, H.A.; Resources, H.A.; Supervision, H.A.; Validation, H.A.; Visualization, H.A.; Writing—original draft, H.A.; Writing—review & editing, N.K.

**Funding:** This research received no external funding.

**Conflicts of Interest:** The authors declare no conflict of interest.

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
