# Peer review of "A Structural Analysis of Economic Processes by the Use of “Energy Content Value”"

_sustainability, doi:10.3390/su11061794_

Round 1

Reviewer 1 Report

I have reviewed before the paper. As of this moment the paper clearly looks much better and it is much easier to follow. I have a few suggestions for improving the paper:

Explain better the main difference between the traditional I-O table and the energy augmented I-O. How would macroeconomic variables like GDP be affected. What are the macroeconomic implications

The examples in Section 4.2 seem relevant, but I would like to see some sharper conclusions and a more tighter connection with the theoretical presentations in the previous sections.

The paper needs to discuss too the limitations and what open questions remain.

Author Response

In Section 3.3, we have accepted the traditional I-O table as is given, without any augmentation or reduction, but projected (transferred) the industrial sectors (including the GDP) onto the energy axis.

We add the following description at the end of section 4.2.
"As mentioned above, our assessment using ECV enables a long-term cost analysis on bypassing  fluctuations and/or drift over decades in the currency value."

We added a description to the conclusion as shown in red.

Reviewer 2 Report

Thank you for the revision. I believe the draft succeeds at outlining the main point of the paper. There are a few minor typos and mis-spelling mistakes to be fixed, but I see no problem with that.

Author Response

We checked and corrected some mistakes.

Reviewer 3 Report

The manuscript presents a theoretical framework to evaluate the overall economic activity by the use of "Energy Content Value". An alternative evaluation indicator based on the "energy" is proposed. It is considered as the most important economic substance and the indispensable requirement for all human activities. The theoretical foundation of the research is based on the issues raised by Nicholas Georgescu-Roegen and the theoretical insight of Frederic Soddy.  

This is a very interesting topic of economic processes analysis in relation to sustainability issues. Taking under consideration the presentation as a whole, I would say that an interesting and substantial work has been done.  

I found this a useful and thoughtful manuscript which makes some very good comments about the evaluation of economic processes about existing and proposes a corresponding structural analysis.

In this revised edition, the authors have covered my main comments regarding (a) the research questions, (b) the theoretical analysis and (c) the original approach to the analysis of the problem.

The manuscript is mostly well written and clearly structured and organized. I recommend the manuscript to be published with some minor improvements.

1.      The titles of the Tables and Figures should be properly located following the format of the journal.

2.      Page 4, Line 147: “in the dimension of”?

3.      Table 2: Yen or yen?

4.      Table 3: Text format.

            5.      Ref. 11: Date is missing

Author Response

1. The titles of the Tables and Figures should be properly located following the format of the journal.

We reviewed our manuscript so as to meet the format.

2. Page 4, Line 147: “in the dimension of”?

The word "Energy" was missing.

3. Table 2: Yen or yen?

We fixed it to "Yen".

4. Table 3: Text format.

Because the table might collapse, the image data is left as it is.

5. Ref. 11: Date is missing

We compensated for the date.

Round 2

Reviewer 1 Report

The authors have adequately responded to all the raised issues.

This manuscript is a resubmission of an earlier submission. The following is a list of the peer review reports and author responses from that submission.

Round 1

Reviewer 1 Report

This is the second time I review the paper. The authors have clearly tried to improve the paper. However, there are still some important issues that are not really addressed.

First, my impression is that the paper tries to achieve to much. The questions it aims at answering to are very general, and deserve a study in themselves.

Second, there a fundamental issue - namely, the authors do not argue convincingly how GDP can be measured by energy content. Although I understand this tendency towards imitating physics, economics deals with human choice and not particles and atoms. As such, simply transplating the physics concepts and methodology cannot be done except some limits.

Third, the empirical section (case study of Japan), altough an improvement, does not really address the issue of the causality between energy and production. This is not an easy to answer question, however the authors should use the recent advances regarding causality in economics.

Reviewer 2 Report

Thank you for re-writing the paper and for replying to my remarks in a sharp way. I am happy to recommend a publication decision for the draft.

I wish the authors the best of luck with their future endeavours.

Reviewer 3 Report

The manuscript presents a theoretical framework to evaluate the overall economic activity by the use of "Energy Content Value". The analysis is based on the issues raised by Nicholas Georgescu-Roegen and the theoretical insight of Frederic Soddy.  

This is a very interesting topic of economic processes analysis in relation to sustainability issues. Taking under consideration the presentation as a whole, I would say that an interesting and substantial work has been done.  

The manuscript is well written and organized. It is suitable for publication and I only have minor comments.

1.      The titles of the sections should follow the format of the journal.

2.      The figures need to be improved.

3.      Fig. 4: horizontal axis should be added

4.      The conclusions could be better presented

            5.      Ref. 24: It should be written in English